# A Critical Look at Omega-3 Supplementation: A Thematic Review

**DOI:** 10.3390/healthcare11233065

**Published:** 2023-11-29

**Authors:** Yamil Liscano, Natalia Sanchez-Palacio

**Affiliations:** 1Grupo de Investigación en Salud Integral (GISI), Departamento Facultad de Salud, Universidad Santiago de Cali, Cali 5183000, Colombia; 2Grupo de Investigación en Promoción de la Salud y Prevención de la Enfermedad, Universidad de Caldas, Manizales 170004, Colombia; natalia.sanchez@ucaldas.edu.co

**Keywords:** omega-3, PUFA (Polyunsaturated Fatty Acids), postpartum depression, supplementation

## Abstract

Postpartum depression (PPD) affects 10–20% of women. Traditional treatments have raised concerns, but omega-3 fatty acids show potential as an alternative. This thematic review, sourced from databases like PubMed and Scopus between 1 February 2023 and 15 March 2023, seeks to delve into the various perspectives on omega-3 supplementation for PPD. The criteria included studies detailing depressive symptoms, social functioning, and neurobiological variables. The review includes research with women showing PPD symptoms, randomized clinical trials, and articles in Spanish, English, and French. Exclusions were studies lacking proper control comparisons and other interventions besides omega-3. Data extraction was performed independently. Two key studies provide contrasting findings on omega-3’s impact on PPD symptoms. In the study comparing DHA supplementation to a placebo, significant differences were not found in the EPDS scale, but differences were observed in the BDI scale. In contrast, another study recorded a significant decrease in depression scores in all dose groups, with reductions of 51.5% in the EPDS scale and 48.8% in the HRSD scale. Other studies, encompassing both prenatal and postpartum periods, underscore the differentiation between prenatal depression and PPD. Despite shared diagnostic criteria, PPD presents unique symptoms like restlessness, emotional lability, and baby-related concerns. It is crucial to address biases and obtain specific results, recommending exclusive PPD-focused studies. This review emphasizes the need for continuous exploration of omega-3’s relationship with PPD to enhance the life quality of pregnant women and their families.

## 1. Introduction

Affective disorders are the second most prevalent mental illnesses globally. Notably, depression disproportionately affects women, often triggered by significant life events such as childbirth. In regions like Colombia and other Latin American countries, the issue of postpartum depression (PPD) is notably underdiagnosed and often not adequately addressed. Tackling this public health concern is imperative given its profound influence on women’s lives and their broader family units [1,2].

During the first year after childbirth, PPD affects between 10% and 20% of women. This disorder can have severe consequences for the mental and emotional health of both the mother and the child, as well as affecting the quality of life of the entire family [1]. Treating depression during pregnancy raises concerns for both mothers and physicians regarding fetal safety [1,2]. Many mothers are reluctant to take medications during gestation [3,4,5]. Selective serotonin reuptake inhibitors (SSRIs) are the standard treatment for prenatal depression and PPD and are widely prescribed during and after pregnancy. Some observational studies, though not all, have pointed to an increased risk of congenital heart malformations or pulmonary hypertension with the use of these medications. SSRIs easily cross the placenta, and the effects of these compounds on the developing brain are unknown [3,4,5,6].

With the emergence of new alternatives for the treatment of PPD, some women choose these new options, such as mindfulness, acupuncture, and supplementation with folic acid and omega-3 [3,4,7,8,9]. Among these, omega-3 supplementation has been shown to be effective in treating depression and has anti-inflammatory and neuroprotective properties [4].

This essential fatty acid is found in foods such as fish and some plants and has been shown to have anti-inflammatory and neuroprotective properties that can improve the mental and emotional health of individuals [10]. Omega-3 fatty acids are long-chain polyunsaturated essential fatty acids that must be obtained through the diet or synthesized from a precursor. During pregnancy, there is a higher risk of omega-3 deficiency, making supplementation advisable [11,12]. Fish oil is considered a promising alternative treatment for mood disorders during pregnancy due to its potential benefits for fetal development and limited side effects. There is evidence suggesting that omega-3 supplementation during pregnancy can help prevent maternal depression. In low-income countries, omega-3 supplementation may be a useful alternative treatment for maternal mood disorders due to limited access to healthcare services [3,4,5,6].

The aim of this research is to conduct a review of the effects of omega-3 supplementation in women with PPD. The significance of this study lies in the thorough review of the existing literature to determine the efficacy and safety of omega-3 supplementation in the treatment of PPD. Equally important, the findings of this work can influence public health and the formulation of policies and strategies to address PPD. This could lead to the implementation of early detection programs, provision of support services, and appropriate treatments. Furthermore, it would drive more research and clinical studies to delve deeper into the relationship between omega-3 supplementation and PPD.

## 2. Materials and Methods

### 2.1. Review Methodology

#### 2.1.1. Definition of Concepts

PPD: It is a depressive disorder that occurs in women after childbirth. It is characterized by a state of persistent sadness and anxiety, fatigue, and loss of interest in previously rewarding activities. Some women may also experience changes in appetite, insomnia or excessive sleep, feelings of guilt or shame, and suicidal thoughts. PPD can have serious consequences for the mental and emotional health of the mother and her child, and seeking treatment to address the symptoms is essential [1,6].

#### 2.1.2. Search Strategy

The searches were conducted in online databases including PubMed, Scopus, Science Direct, Biomed Central, Cochrane, Google Scholar, LILACS, Web of Science, and Virtual Health Library (VHL) from 1 February 2023 to 15 March 2023. The search strategy used in PubMed is described in Appendix A and adapted as needed for implementation in other databases. The following tools were used for data storage and selection: Zotero version 6.0.27 (accessed on 20 May 2023) and Rayyan 1.3.0 (accessed on 20 May 2023).

#### 2.1.3. Inclusion and Exclusion Criteria

This study considered research meeting the following inclusion criteria: studies reporting on depressive symptoms, social functioning, and neurobiological outcomes; inclusion of women who had recently given birth and exhibited symptoms of postpartum depression (PPD); articles that were fully accessible; randomized clinical trials (RCTs); and articles written in Spanish, English, or French. On the other hand, the exclusion criteria encompassed research that combined omega-3 intervention with other therapeutic approaches, such as cognitive behavioral therapy. Furthermore, studies that centered on women with a history of depression before pregnancy or those presenting with other psychiatric conditions were also excluded. In addition, studies with a duration of less than 4 weeks were excluded due to the potential delayed effects of omega-3 on PPD.

#### 2.1.4. Selection and Data Extraction

Two authors (Y.L. and N.S.-P.) independently performed data extraction and allocation on pre-defined data collection forms and resolved any discrepancies via consensus. The data detailing the trial (first author, year of publication), participant (age, sex, and composition [EPA vs. EPA+DHA with doses]), and study design characteristics (study type, intervention, study duration, adverse effects) were noted. Any discrepancies in article selection and data extraction were resolved via discussion.

## 3. Results

### 3.1. PPD: A Particular Variant of Depression

Perinatal depression, a mood disorder characterized by persistent feelings of sadness and loss of interest, can occur during or after pregnancy. It affects the mental and physical well-being of individuals and can cause significant emotional, relational, and physical problems. While it is normal for new mothers to experience mild mood changes and anxiety, known as “baby blues”, immediately after childbirth, persistent or severe symptoms beyond two weeks may indicate PPD [7,8]. According to the DSM-5, PPD is characterized by the presence of a major depressive episode occurring within 4 to 6 weeks after childbirth and can be symptomatically defined by exceeding a certain threshold on a screening measure, such as the Edinburgh Postnatal Depression Scale (EPDS). Symptoms may include deep sadness, lack of interest or connection with the baby, changes in appetite and sleep, fatigue, irritability, anxiety, feelings of guilt or worthlessness, and recurrent thoughts of harming oneself or the baby [2]. According to recent estimates, 7% of women experience a severe depressive episode in the first three months after childbirth, and the prevalence increases to 20% if mild depressive episodes are also included [9].

It is important to note that there is a difference between postpartum sadness or minor puerperal depression (baby blues) and PPD proper. While postpartum sadness typically lasts for 2 to 3 days, or up to 2 weeks, and is relatively mild, PPD persists for more than 2 weeks, is disabling, and interferes with daily activities [10,11,12].

Untreated PPD can have serious consequences, such as difficulties in the bonding process, breastfeeding problems, and an increased risk of self-harm or harm to the baby. Furthermore, children affected by mothers with PPD may develop issues such as lower cognitive functioning, behavioral inhibition, emotional maladjustment, violent behavior, externalizing disorders, and psychiatric and medical disorders during adolescence [2,13]. This type of depression often goes unnoticed and is not adequately treated, making it essential for healthcare professionals to perform active assessments and address this condition [3,11].

### 3.2. Epidemiology and Risk Factors of PPD

PPD is influenced by several risk factors that increase the likelihood of its development in women after childbirth. Some common risk factors include a history of previous depression or mood disorders, family history of depression, stress during pregnancy or stressful life events, lack of social support, relationship issues or conflict with the partner, financial difficulties, and physical health problems or complications during pregnancy or childbirth [3,14]. Additionally, women who have experienced multiple adverse life events, including childhood or adult sexual abuse, are at higher risk of PPD, being three times more likely to experience it compared to those who have not experienced any adverse events in life [8]. According to Zhao and Zang, 2020 [15], women who experience complications during pregnancy or childbirth have an increased risk of developing PPD. This review suggests that it is essential to identify at-risk women and provide them with appropriate support to prevent or treat PPD.

Maternal depression is recognized as a public health problem worldwide. In high-income countries, between 10% and 20% of women experience depression during pregnancy or in the first year after childbirth. However, it is important to consider that prevalence may vary by country, culture, and the methods used to assess and diagnose PPD [3,14,16].

In low- and middle-income countries, the prevalence of depression during pregnancy ranges from 13% to 30%, while in the postpartum period, it varies between 13.8% and 32.9%. In Brazil, among women living in metropolitan areas, the prevalence of depressive symptoms in the postpartum period ranges from 20.7% to 39.4%. In Colombia, the prevalence of PPD was found to be 12.9%, higher in urban areas (15.1%) than in rural areas (6.8%), and among women aged 20 to 34 years (13.4%). Geographically, the departments of Amazonas and Guainía have a lower prevalence of PPD (3.1% and 3.5%, respectively), while in the city of Bogotá and the department of Quindío, the prevalence was 18.1% and 22.1%, respectively [17,18].

A study conducted by Latorre-Latorre et al., 2006, in women from Bucaramanga and Floridablanca, Colombia, found a prevalence of 24.6% for depression and 25.8% for gestational anxiety. Depression was associated with a family history of depression, the presence of anxiety, and alcohol consumption. Having two sources of income (partner and family) was found to be a protective factor. Anxiety was associated with the presence of depression, psychological violence, and lack of confidence in the partner [19,20]. Research by Loaiza and Sánchez, 2019, at the Versalles Clinic in the department of Caldas found that the prevalence of PPD in the evaluated population was 26%. Out of the 39 postpartum women assessed, 10 exhibited symptoms of PPD, highlighting the need for early evaluation and multidisciplinary intervention to prevent psychological issues and ensure the well-being of mothers and their newborns. There were no statistically significant relationships between age and PPD, but it was determined that PPD is more common in adult postpartum women compared to adolescents. Most women with PPD had a normal delivery, and the incidence of PPD was lower than in previous studies [21].

In 2013, out of every 1000 female adolescents aged 15 to 19 in the United States, there were 26.5 births, totaling 273,105 babies born to teenage mothers. Despite the decline in the annual rate of teenage pregnancies in the last decade, the United States still leads in teenage pregnancies among developed countries. The prevalence of depression in teenage mothers ranges from 14% to 53%, which is higher than the 6.9% to 16.7% observed in adult mothers. Compared to adult mothers, teenage mothers are more likely to have used illicit drugs before pregnancy, smoked during pregnancy, neglected prenatal health recommendations, and experienced physical abuse from the baby’s father. Finally, compared to teenagers in the prepartum period, teenage mothers are three times more likely to attempt suicide (20% vs. 6.3%) [22].

In the African continent, the reported prevalence of PPD in countries like Nigeria and Uganda is 8.8% and 27.1%, respectively [23,24]. According to the report by Okunola et al., 2021 [23], from southwestern Nigeria, having a history of depression, an unplanned pregnancy, and lack of social support are some of the main predictors of PPD. On the other hand, the study by Athuaire et al., 2021 [24], in Uganda found that five variables showed statistically significant predictive values for PPD among the mothers surveyed in southwestern Uganda. These factors include having a complicated childbirth, having a baby who was hospitalized in the last six weeks, having breastfeeding problems, having baby sleep problems, and having a baby who cries excessively [23,24].

In the Asian continent, in Nepal and Bangladesh, the prevalence of PPD was 14.7% and 34%, respectively [25,26]. The study conducted by Pradhananga et al., 2020 [25], in Nepal revealed that the prevalence of PPD was 14.7%. This figure was considerably higher compared to the results obtained in several developed countries such as Norway (10.1%), Sweden (12.5%), and Canada (8.46%). However, compared to developing countries such as India (22%), Pakistan (28.8%), Indonesia (22.35%), and Argentina (18.6%), the prevalence of PPD was lower. These findings suggest, according to the authors, that PPD is more common in developing countries, especially in Asian nations. This disparity could be due to the differential access to quality healthcare services available to women in developed countries, such as family planning, reproductive health programs, prenatal and postnatal care services, safe childbirth, promotion of maternal nutrition, child care, among others. In the present study, PPD was more common in older women (ages 36 to 45 years), which could be attributed to older women having difficulties coping with the consequences of pregnancy compared to younger mothers. Employed postpartum mothers may have an additional workload compared to homemakers, which could be a contributing factor to PPD [25].

Tasnim et al., 2021 [26], in Bangladesh found that approximately one-fourth of pregnancies in the sample were unwanted, with 14.3% reported as mistimed and 10.3% as unwanted. There was a significant association between unwanted pregnancy and an increased risk of developing PPD. Women with unwanted pregnancies may experience higher stress and inadequate preparation for pregnancy, childbirth, and breastfeeding, which can lead to anxiety, helplessness, and difficulties in coping with the postpartum period. Additionally, the experience of violence from spouses or in-laws, which is common in Bangladeshi society, may contribute to the relationship between unwanted pregnancy and PPD [26].

### 3.3. Diagnosis and Conventional Treatment of PPD

PPD and non-perinatal major depression share the same diagnostic criteria, including a combination of symptoms such as depressed mood, loss of interest, sleep and appetite disturbances, difficulty concentrating, fatigue, and suicidal thoughts. These symptoms must be present for at least two weeks and cause significant distress or impairment in functioning. In addition to these symptoms, PPD also includes mood lability, anxiety, irritability, and obsessive concerns or worries about the health and safety of the baby [27,28].

Detecting PPD involves a sensitive clinical inquiry about mood during postpartum follow-up visits with obstetric or primary care personnel. Various screening tools have been developed to aid in identifying PPD, such as the EPDS and the Patient Health Questionnaire (PHQ), which assesses feelings experienced in the last 2 weeks. Both the full PHQ and PHQ-9 have effectively identified patients with and without major depression in primary and obstetric care, demonstrating high sensitivity and specificity. Another measure is the Center for Epidemiologic Studies of Depression (CES-D) instrument, a 20-item questionnaire where scores ≥ 16 indicate depression based on symptoms experienced in the last 7 days. CES-D scores remain stable during the first postpartum year and are associated with the diagnosis of depressive disorders. Lastly, the Postpartum Depression Screening Scale (PDSS) is another tool that evolved from qualitative interviews to explore the maternal experience after childbirth. The PDSS has been effectively used in telephone screenings and in Spanish-speaking and Native American communities [28,29].

The sensitivity and specificity combined for the EPDS are maximized with a cutoff value of EPDS ≥ 13. An EPDS score of ≥13 is related to a Hamilton Rating Scale for Depression (HRSD) score of ≥20, suggesting a high probability of a major depressive episode [28,29,30]. Furthermore, the accuracy does not significantly differ based on reference standards or participant characteristics, including whether the EPDS is administered during pregnancy or in the postpartum period. However, there is controversy regarding the routine use of screening, as it requires appropriate treatment and follow-up systems. Different guidelines recommend different screening approaches, and clinical assessment is considered the gold standard for diagnosis [29,30,31].

Another issue to consider in the diagnosis of PPD is the differential diagnosis, as it is important to distinguish PPD from other mental disorders and medical conditions. Medical conditions such as thyroid dysfunction and anemia should be ruled out as they can cause depressive symptoms. Postpartum “baby blues,” a mild and transient syndrome of low mood, crying, and mild irritability, can be distinguished from PPD by the severity and persistence of symptoms. Adjustment disorders, which involve emotional or behavioral symptoms in response to a stressor, are often present in the postpartum period. Depressive disorders can coexist with anxiety, obsessive-compulsive, and trauma-related disorders. It is crucial to assess the history of depression, bipolar disorder, and postpartum psychosis, as they require different treatment approaches. Postpartum psychosis, a manifestation of bipolar mood disorder, is a psychiatric emergency that requires immediate hospitalization due to its rapid fluctuating course and the potential for harm to the mother or baby [28,29,31].

Regarding treatment, PPD can be addressed using different treatment approaches. One of these approaches is psychotherapy, such as supportive therapy, cognitive behavioral therapy, and interpersonal therapy [4]. Supportive therapy provides a safe space for the mother to express her feelings and concerns, where a mental health professional offers emotional support and coping strategies. Cognitive behavioral therapy focuses on changing the negative thought patterns and dysfunctional behaviors associated with PPD, helping the mother identify and replace negative thoughts with more realistic and healthier ones, as well as developing effective coping skills. Interpersonal therapy aims to improve the mother’s communication skills and interpersonal relationships, addressing relationship issues, role changes, and social difficulties that may contribute to PPD [4,27,32,33].

In moderate to severe cases of PPD, antidepressant medications can be prescribed, with selective serotonin reuptake inhibitors (SSRIs) being the most commonly used during breastfeeding due to their lower side effects on the baby [1,5,34,35]. The medications include SSRIs such as sertraline (Zoloft), fluoxetine (Prozac), escitalopram (Lexapro), and paroxetine (Paxil) [36,37]. Brexanolone is a medication specifically designed to treat PPD. It works by targeting the GABA-A receptor in the brain, which helps regulate mood and anxiety. By enhancing the activity of this receptor, Brexanolone can help alleviate symptoms of PPD, such as sadness, anxiety, and irritability [38]. These medications work by selectively inhibiting the reuptake of serotonin in the brain. Another class of medication is serotonin–norepinephrine reuptake inhibitors (SNRIs), such as venlafaxine (Effexor) and duloxetine (Cymbalta), which inhibit the reuptake of both serotonin and norepinephrine. Bupropion (Wellbutrin) belongs to the class of norepinephrine–dopamine reuptake inhibitors (NDRIs), while amitriptyline (Elavil), nortriptyline (Pamelor), and mirtazapine (Remeron) are tricyclic antidepressants (TCAs) that affect serotonin and norepinephrine levels. Each class of medication has specific mechanisms of action and may be prescribed based on individual needs and response to treatment [1,36,38,39,40,41].

However, it is essential to carefully evaluate the balance between the benefits and risks of medication during pregnancy and breastfeeding. In some cases, hormonal treatments, such as hormone replacement therapy, may be considered for women with PPD symptoms related to significant hormonal changes [1,5,34,35].

During the postnatal period, women suffering from depression may face the difficult decision of receiving treatment with antidepressants while breastfeeding their babies. It is important to recognize that antidepressant medications can have adverse effects on both the mother and the infant during breastfeeding. Some antidepressants can pass into breast milk in significant amounts and be absorbed by the nursing baby during feeding [36,40,42,43]. This raises concerns about the potential adverse effects these medications may have on the development and well-being of the baby [34,40,41]. Among antidepressants, sertraline is one of the safest for breastfeeding women. It is generally recommended that women who are already taking sertraline continue breastfeeding while on the medication, starting with low doses and gradually increasing as needed. It is important to closely monitor the newborn for any adverse effects, especially in premature or low-birth-weight babies. Avoiding breastfeeding during the time of the highest concentration of antidepressants in breast milk can help minimize the baby’s exposure. Switching to sertraline from another effective treatment should only be considered after a thorough evaluation of the risks and benefits, with careful monitoring of the baby [44].

### 3.4. Omega-3 as an Alternative in the Treatment of PPD

Omega-3 fatty acids are polyunsaturated and are characterized by having more than one double bond. Their name comes from the location of the first double bond on the third carbon atom, counting from the methyl end of the fatty acid. These fatty acids are widely used as non-vitamin supplements in the United States, both by children and adults. Various research studies are being conducted to determine whether omega-3 plays a significant role in mental health and diseases.

Some epidemiological studies suggest that individuals consuming a diet rich in omega-3 have a lower risk of developing major depression, prenatal depression, and bipolar depression. However, compared to other countries, the Western diet has a low content of omega-3 fatty acids and a high content of omega-6 fatty acids, which may be associated with inflammation [18,45,46]. It has been suggested that omega-3 fatty acids may have positive effects on depression by interacting with serotonin and dopamine transmission. Additionally, an anti-inflammatory effect and regulation of the immune response have been observed [46,47].

The relationship between depression and diet has been the subject of research, especially concerning the intake of omega-3 fatty acids. Epidemiological studies have yielded contradictory results; some suggest an inverse association between omega-3 consumption and depressive symptoms, while others have not found a significant relationship. Similarly, experimental studies have produced inconsistent findings. While some indicate a positive effect of omega-3 in reducing depressive symptoms, others have not found significant benefits. Methodological limitations and differences in participant characteristics and omega-3 regimens used may explain these discrepancies [48,49,50,51,52].

### 3.5. Evidence from Clinical Trials in the Treatment of PPD

In the process of selecting relevant manuscripts for randomized clinical trials focused on alternative treatments for PPD using omega-3, several filtering stages were employed. Initially, a total of 1335 manuscripts were identified from various registers. However, not all of these were unique; 198 duplicate records were identified and subsequently removed. The next phase involved a rigorous screening process, which further reduced the pool to only eight manuscripts that passed the inclusion and exclusion criteria, notably including perinatal studies. Out of these eight, six were perinatal studies and not strictly postpartum, leading to their exclusion. As a result, only two manuscripts were found to be strictly related to randomized clinical trials of postpartum depression (see Figure 1).

Table 1 provides information on two different studies related to omega-3 fatty acid supplementation in women with PPD. In the study by Llorente et al., (2003) [53], a double-blind, randomized, placebo-controlled trial was conducted. Pregnant women who planned to exclusively breastfeed their babies for at least 4 months were enrolled in the study. Participants had to meet specific inclusion and exclusion criteria, which included being between 18 and 42 years old, not suffering from any chronic disease, not taking dietary supplements other than vitamins, not smoking, and not having been pregnant more than five times.

In the study (Freeman et al., 2006 [54]), a randomized dose-ranging pilot trial was conducted, where participants were randomly assigned to one of three groups with different doses of omega-3 fatty acids or a placebo group. Participants were women who had given birth within the last 6 months and met the diagnostic criteria for PPD. The age range was set from 15 to 45 years (with a mean age of 31 years), and women were required to meet the diagnostic criteria for a major depressive episode with onset within 1 month after childbirth and have a specific score on the depression scales used (the HRSD and EPDS). The exclusion criteria included participants with prior intolerance or allergy to omega-3 fatty acids, current use of antidepressant medications, psychotic symptoms, history of mania/hypomania, and active suicidal ideation.

Regarding the intervention with omega-3 in the clinical trials, the study by Llorente et al., 2003 [53], used an intervention of 200 mg of DHA in pregnant women. Several scales were used to assess depression symptoms in the pregnant women, including the Beck Depression Inventory (BDI), the EPDS, and the Structured Clinical Interview for Axis I Disorders from the DSM. The results showed that no significant differences were found between the DHA and placebo groups in the EPDS scale, but significant differences were observed in the BDI scale. The intervention lasted for 4 months, and no specific side effects were mentioned.

On the other hand, the study by Freeman et al., 2006 [54], used a supplement of omega-3 fatty acids with an EPA:DHA ratio of 1.5:1 in three different dose groups: 0.5 g/day, 1.4 g/day, or 2.8 g/day. The study initially involved 21 women, and at the end, 16 women remained in the study. The EPDS and the HRSD were used to assess depression symptoms. The results showed a significant decrease in depression scores in both groups, with a reduction of 51.5% in the EPDS scale and 48.8% in the HRSD scale. The intervention lasted for 8 weeks, and no specific side effects were mentioned.

As for the findings, authors’ conclusions, and limitations of both studies, the following were found:

In the study by Llorente et al., 2003 [53], no significant differences were found in PPD symptoms or information processing measures between women receiving DHA supplements and those receiving a placebo. Although the plasma DHA content increased in the supplementation group, the results suggest that the amount of DHA used in this study did not have a significant effect on depression symptoms or cognitive processing. It is suggested that future investigations consider higher doses of DHA or combinations of essential fatty acids to evaluate potentially more pronounced effects. It was mentioned that populations with a high intake of omega-3 have lower rates of major depression and PPD. It was suggested that plasma DHA concentration might be related to serotonin metabolite concentration in the central nervous system, which could explain the relationship between omega-3 consumption and a reduction in depression symptoms. However, more studies are needed to confirm these findings and determine the optimal dose of omega-3. The study limitations included the focus on a specific sample of women who planned to exclusively breastfeed their babies for at least 4 months, limiting the generalization of the results to other populations. Additionally, other unmeasured factors, such as women’s diet, could have influenced the study results. Factors like stress or sleep quality, also unmeasured, could have influenced the results.

In the study by Freeman et al., 2006 [54], it was found that a combination of EPA and DHA was beneficial and well tolerated for women with PPD. Depressive symptoms significantly improved in all dose groups from the baseline, and there were no significant differences in treatment response between dose groups. The limitations of the study include a small sample size, lack of a placebo group, and uneven randomization in the groups. Additionally, no biochemical measures were performed to evaluate essential fatty acid plasma levels, which could have helped determine which patients would benefit most from omega-3 supplementation.

### 3.6. Other Studies on Postpartum Omega-3 Consumption with the Addition of Prenatal Consumption

While there are other studies addressing PPD, it is essential to note that these studies also include the prenatal period. Therefore, it should be considered that PPD and prenatal depression are distinct clinical entities with different characteristics, risk factors, and consequences. By including studies that cover both periods, there is a risk of diluting specific findings related to PPD and hindering the acquisition of accurate information about this condition [55].

Prenatal depression refers to depressive symptoms being experienced during pregnancy, while PPD focuses on symptoms that develop after childbirth. These two periods have significant differences in terms of the target population, study duration, symptom assessment, outcomes and consequences, interventions and treatments, as well as the biological, social, and environmental factors involved [34,56,57,58,59]. During pregnancy, hormonal changes occur that can have a significant impact on women’s mental and physical health. These hormonal changes during pregnancy may be associated with psychiatric disorders and can affect hormonal balance, such as sex steroids and cortisol [60]. On the other hand, hormonal changes during childbirth and postpartum, such as the drastic decrease in progesterone and estradiol, along with the production of endorphins and adaptation to lower estradiol levels, may contribute to the mood changes and emotional vulnerability experienced during this period (see Table 2). Furthermore, fluctuations in thyroid hormones can also play a role in the mother’s mental health [56].

The identification of nutritional and biochemical markers for the diagnosis of PPD has gained recent attention. Several studies have revealed correlations between the severity of depressive symptoms and decreased serum concentrations of zinc, vitamin D, and omega-3 in patients with PPD. These findings could lead to the development of more effective preventive and therapeutic measures for PPD, benefiting both the mothers and children affected by this condition [60,61]. Hoge et al., 2019 [61], found that a higher omega-6 to omega-3 ratio was associated with a greater risk of PPD. The balance between these fatty acids is important to understand their influence on health. The Arachidonic acid/Eicosapentaenoic acid ratio in cell membranes is related to inflammatory processes during depression, which have been linked to serotonin depletion and the accumulation of neurotoxic quinolinic acid metabolites. Additionally, the study highlighted the importance of the omega-3 index in red blood cell membranes, where women with an index below 5% were approximately five times more likely to experience PPD compared to those with an index of 5% or higher.

Although PPD and non-perinatal major depressive disorder share similar diagnostic criteria, PPD is distinguished by symptoms such as restlessness, lethargy, emotional lability, and baby-related concerns. Furthermore, PPD is often accompanied by anxiety, recurrent thoughts, and panic attacks, with anxiety disorders being common in women experiencing PPD. It is estimated that around 7% of women suffer from major depression in the first three months after childbirth, a figure that increases to 20% when considering episodes of minor depression [9,56].

To obtain more accurate results about PPD, it is important to address various biases and avoid confusion in the data interpretation by merging results from studies that include both prenatal depression and PPD. This means considering and analyzing each stage separately, allowing for a clearer and more specific understanding of the factors related to PPD. By addressing these biases, a better understanding of the unique nature and challenges associated with PPD can be obtained. For instance, selection bias may arise when heterogeneous participants are included in perinatal clinical trials, making it difficult to generalize the results. Information bias may occur if perinatal studies do not collect comprehensive data on PPD, leading to a lack of detailed information about symptoms and treatment. Finally, time bias emphasizes the importance of specifically assessing the effects of omega-3 on PPD, as evaluating it throughout the entire perinatal period may hinder the identification of specific effects on this disorder. To address these biases and obtain more specific results, it is recommended to include studies focused exclusively on PPD and evaluate the effects of omega-3 in this context [62,63,64].

Despite the aforementioned notes, it is worth comparing the results of the two postpartum reviews with their counterparts from perinatal studies, as evidenced in Table 3. The published authors belong to the USA, the Netherlands, Australia, Brazil, and Iran. Although these studies share a similar focus, there are differences in the types of participants and the ages of the participants in each study.

In the study by Freeman et al., 2008 (USA) [65], pregnant women were recruited after 12 weeks of gestation, as it was considered a period when morning sickness decreased. These women had an average age of 30 years. On the other hand, Rees et al., 2008 (Australia) [66], included women in the perinatal period, meaning both pregnant women and those who had recently given birth were included. In this study, they were divided into two groups: one receiving fish oil and the other receiving a placebo. The fish oil group had an average age of 31.2 years, while the placebo group had an average age of 34.5 years.

In the study by Doornbos et al., 2009 (The Netherlands) [67], the age range of the recruited pregnant women in Groningen was not specified. In the study by Mozurkewich et al., 2013 (USA) [68], the participants were pregnant women receiving prenatal care in two health systems in southeastern Michigan. However, the average age of these participants was not mentioned.

In the study by Vaz et al., 2017 (Brazil) [18], pregnant women attending a prenatal clinic in the city of Rio de Janeiro were recruited. These women were between 20 to 26 weeks pregnant and were followed up until 4 to 6 weeks after delivery. The age range of these participants ranged from 18 to 40 years.

Finally, in the study by Farshbaf-Khalili et al., 2017 (Iran) [69], pregnant women attending primary health care centers in Tabriz, Iran, were recruited. The sample was stratified based on the number of pregnancies, including women who had been pregnant once, twice, or more. The placebo group had an average age of 26.9 years, while the fish oil group had a slightly lower average age of 25.9 years.

The study by Freeman et al., 2008 (USA) [65], included 59 pregnant women who were randomly assigned to receive either omega-3 fatty acids or a placebo. Additionally, they received six sessions of supportive psychotherapy during the trial. The EPDS and the Hamilton Rating Scale for Depression (HAM-D) were used to assess depression, along with the Clinical Global Impression (CGI) to evaluate disease severity and treatment progress. The intervention duration ranged from 12 to 32 weeks of gestation or up to six months after childbirth.

In the study by Rees et al., 2008 (Australia) [66], 26 pregnant women were randomly assigned to either a treatment group receiving fish oil or a placebo group. Several scales were used, including the EPDS, the HRSD, and the Montgomery–Åsberg Depression Rating Scale (MADRS). The intervention duration was six weeks from the start of treatment.

In the study by Doornbos et al., 2009 (The Netherlands) [67], 119 pregnant women were randomly assigned to three different groups to take either soybean oil (placebo), DHA, or DHA + AA daily from the 14th to 20th week of pregnancy until three months after childbirth. The scale used to measure depression was the EPDS. The intervention duration was from the 16th week of pregnancy until three months after childbirth.

Mozurkewich et al., 2013 (USA) [68], randomly assigned 118 pregnant women to one of three intervention groups: supplementation with EPA-rich fish oil, supplementation with DHA-rich fish oil, or soybean oil placebo. The EPDS, the BDI, and a structured clinical interview called MINI were used to measure depression. The intervention started between the 12th and 20th weeks of gestation, with follow-up until the sixth or eighth week postpartum.

Vaz et al., 2017 (Brazil) [18], randomly assigned 60 pregnant women to an intervention group receiving fish oil supplements or a control group receiving placebo. The EPDS was used to measure depression at various stages of pregnancy and early postpartum. The intervention lasted from the 20th week of gestation until six weeks after childbirth.

In the study by Farshbaf-Khalili et al., 2017 (Iran) [69], fish oil capsules or placebo were administered to 150 pregnant women. Depression was measured using the EPDS. The intervention lasted 10 weeks and started in the 16th to 20th week of gestation, with postnatal depression evaluated between 30 and 45 days after childbirth.

The most commonly used scales to measure depression in these studies were the EPDS, HAM-D, BDI, and MADRS. These scales were administered at different times during pregnancy and postpartum, depending on the study. The intervention durations also varied, ranging from six weeks to three months after childbirth. The EPDS is a useful tool for detecting postpartum depression, but its validity and utility may vary depending on the population and purpose of use. It is noted that the validity of the EPDS should be interpreted in relation to its intended use. According to Santos et al., 2007 [70], as a screening instrument, the EPDS is suitable using a cutoff point ≥ 10, especially in selected populations of mothers at high risk of PPD. However, for diagnosis, a cutoff point ≥ 13 will be appropriate only if used in high-risk populations [70]. Shrestha et al., 2016 [30], in their study, indicate that local versions of the EPDS used in low- and middle-income countries show deficiencies in translation, cultural adaptation, and empirical validation. These deficiencies impact the accuracy of the scales in identifying real cases of perinatal mental disorders in the general population of women. Therefore, it is recommended to follow a systematic approach that complies with the established steps in the study for the translation, cultural adaptation, and empirical validation of local versions of the EPDS. Regarding the use of this scale for both PPD and depression during pregnancy, Levis et al., 2020 [29], mention that using a cutoff score of 11 or higher on the EPDS provides the best combination of sensitivity and specificity in detecting depression. This means that with a score of 11 or higher, the EPDS correctly identified 81% of women with depression and correctly excluded 88% of women who did not have it. For commonly used cutoff scores of 10 or higher and 13 or higher, the EPDS had a sensitivity of 85% and 66%, respectively, and a specificity of 84% and 95%. Furthermore, the results showed that the screening accuracy was not significantly affected by the cutoffs used or by women’s characteristics, such as whether they were pregnant or in the postpartum period. This indicates that physicians can use a cutoff score of 11 or higher on the EPDS for depression screening. Additionally, there is an online tool that provides further information on screening results based on the findings of this study.

On the other hand, the study by Conradt et al., 2012 [71], suggests caution when using the Beck Depression Inventory-II (BDI-II) to identify PPD, and particular attention should be paid to cognitive/affective symptoms. It is also recommended to develop specific norms for postpartum samples in order to increase the sensitivity and specificity of the BDI-II. Instead of solely focusing on the total score, clinicians could assign weights to certain symptoms when considering depression in postpartum women, which could improve diagnostic accuracy. Finally, the study by Ukatu et al., 2018 [31], provides valuable insights regarding the scales used for PPD, concluding that the diagnostic capacity of PPD screening tools varies due to different factors such as the methodologies used and populations studied. Given the higher prevalence of PPD among patients with low socioeconomic status, obese patients, teenage mothers, racial and ethnic minorities, and immigrant women, it is important to extend research efforts to these populations, and the accuracy of detection methods should be analyzed within the context of this diversity, according to the authors. There is no recommended tool as the most effective, and a greater understanding of cultural and contextual differences is needed. Additionally, initial symptoms may not be just sadness but also include insomnia, anxiety, irritability, and confusion, posing challenges for early detection. Lastly, the authors recommend further research to enhance the identification of women at risk. This includes evaluating the optimal cutoff point of individual tools, determining the best scoring method, identifying the most suitable timing for screening, and finding the most effective combination of tools [31].

Regarding the limitations observed in all the studies, including Llorente et al., 2003 [53], and Freeman et al., 2006 [54], it is evident that they all have small sample sizes, which limits their ability to detect significant differences and generalize the results. Furthermore, several studies mention the need for future research with larger samples and higher doses of omega-3 to draw more robust conclusions. Some studies mention ethical limitations that affect the assignment of placebo groups in certain periods, while others highlight issues related to the duration of treatment and the doses used. In the systematic review with meta-analysis presented by Suradom et al., 2021 [51], on PPD and omega-3 supplementation, the authors acknowledge several limitations of their study. These include the small sample size in most of the included trials, possible inaccuracies in the timing of the initiation of n-3 PUFA supplementation, the use of the “shared group allocation” method in some trials, and a discrepancy in the definition of perinatal depression used in the review compared to the DSM-5 criteria. They conclude that there is insufficient evidence to determine the efficacy of n-3 PUFA supplementation for perinatal depression and emphasize that perinatal women should be informed that n-3 PUFA supplementation has no role in the prevention and treatment of perinatal depression.

Likewise, in this review, we highlight the importance of adopting a reflective and cautious approach when interpreting the available results on the efficacy of n-3 PUFA supplementation for PPD. While the current findings do not support the use of these supplements for treatment, it is crucial to recognize the limitations of existing studies, such as small sample sizes and discrepancies in the definitions and measurements used. Clearly, further research with larger samples and higher doses of omega-3 is needed to draw more robust and reliable conclusions. Additionally, it is essential to consider the validity and utility of the scales used to measure PPD, as well as conduct separate studies to analyze the effects of prenatal and postnatal supplementation. Despite omega-3 supplementation not showing a significant effect on the prevention and treatment of PPD, we should not completely disregard the potential benefits it may have during pregnancy. It has been observed that omega-3 supplementation can have positive effects in reducing the risk of complications such as premature birth, perinatal death, and complications related to low birth weight [51,60,61,72]. In addition to the above, the study by Firouzabadi et al., 2022 [73], shows that omega-3 supplementation during pregnancy can have beneficial effects against preeclampsia and improve anthropometric measures, the immune system, and visual activity in babies, as well as reduce cardiometabolic risk factors in pregnant women.

Based on the evidence previously presented, it can be said that the study of non-pharmacological therapeutic interventions for the treatment of postpartum and perinatal depression is a growing field of interest. The administration of omega-3 fatty acids, in particular, has caught the attention of the scientific community due to its potential effect on mood and neurological functions [50,52,74]. The diversity in the contexts and objectives of the presented studies, as well as the different doses and combinations of EPA and DHA used, display the complexity of addressing this issue. Although all studies had the central objective of exploring the therapeutic potential of omega-3 fatty acids, discrepancies in the results suggest the need to standardize and refine research protocols.

It is relevant to highlight the distinction between postpartum and perinatal depression. Each of these categories has specific characteristics that might influence the response to treatments [1,6,75]. While postpartum depression presents after childbirth, perinatal can manifest during both the prenatal and postnatal periods. This temporal difference could influence the severity of the symptoms, underlying causes, and consequently, the response to omega-3 treatment [55,56,76,77,78,79,80,81,82,83,84,85,86].

The variability in doses and combinations of EPA and DHA in different studies suggests that an optimal range for treatment has not yet been identified. Dosage is a key factor, as an insufficient amount might not yield a therapeutic effect, while an excessive amount could introduce adverse effects or offer no additional benefits. The mixed results presented, from the lack of significant differences in the Llorente et al. 2003 study [53] to the observed symptom decrease in the Freeman et al. 2006 study [54], point to the possibility that there might be uncontrolled or unidentified confounding factors in the studies. Despite the promising signals that omega-3 might be beneficial for treating postpartum, it is evident that the current evidence is insufficient to make firm recommendations. Future clinical trials should be designed with greater methodological rigor, considering standardization in the doses and combinations of EPA and DHA, and taking into account the differences between postpartum and perinatal depression. It is imperative that these studies have adequate sample sizes, appropriate control groups, and longitudinal follow-up periods to determine with certainty the efficacy and safety of omega-3 as a therapeutic intervention.

## 4. Conclusions

The analysis of relevant studies on omega-3 supplementation in women with PPD has shown varied results regarding its efficacy. While one study found significant differences in the depression scale used, another study observed a notable reduction in depression rates across all scales used. However, it is crucial to consider certain limitations of these studies. These limitations may impact the generalization of the findings and the ability to detect significant differences between intervention groups.There is a need for continued research with robust designs, larger samples, and differentiated approaches to PPD. These studies should not only evaluate the effects of omega-3 supplementation but also consider other biological, social, and environmental factors involved in PPD. By expanding our understanding of this condition, we can provide more effective support to women experiencing PPD, thereby promoting their mental and physical well-being as well as the well-being of their family environment.

## Figures and Tables

**Figure 1 healthcare-11-03065-f001:**
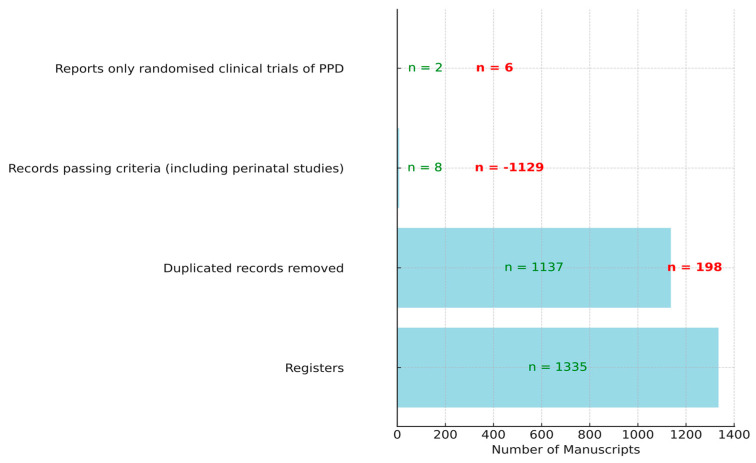
Selection process of manuscripts for randomized clinical trials on alternative treatment for PPD with omega-3. In red, the values of manuscripts that did not pass the filter, and in green, those that passed the filter.

**Table 1 healthcare-11-03065-t001:** Characteristics of clinical trials comparing omega-3 treatment with or without placebo group in women with PPD.

Author	Study Title	Design	Participant Type	Age	Inclusion Criteria	Exclusion Criteria	
Llorente et al., 2003 [53]	Effect of maternal docosahexaenoic acid supplementation on postpartum depression and information processing	Double-blind, randomized, placebo-controlled trial	Pregnant women who planned to exclusively breastfeed their babies for at least 4 months	18–42 years	- Pregnant women aged 18–42 years. - No chronic medical condition. - Not taking dietary supplements other than vitamins. - Non-smokers. - Not pregnant more than 5 times.	Not mentioned.	
Freeman et al., 2006 [54]	Randomized dose-ranging pilot trial of omega-3 fatty acids for postpartum depression	Randomized dose-ranging pilot trial	Women who had given birth in the last 6 months and met criteria for PPD	15–45 years	Women aged 15–45 years who met criteria for a major depressive episode with onset within 1 month after delivery and scored 15 or more on the HRSD or 9 or more on the EPDS.	Intolerance/allergy to omega-3 fatty acids, current use of an antidepressant medication, psychotic symptoms, history of mania/hypomania, and active suicidal ideation.	
**Author**	**Intervention**	**N**	**Scale Used**		**Results**	**Intervention Duration**	**Side Effects**
Llorente et al., 2003 [53]	200 mg DHA	101	1. Beck Depression Inventory (BDI), used to assess depression symptoms in pregnant women longitudinally.		No significant differences were found between the DHA and placebo groups in the EPDS scale, but in the BDI scale.	4 months	Not mentioned
			2. Edinburgh Postnatal Depression Scale (EPDS) and the Structured Clinical Interview for Axis I Disorders from the Diagnostic and Statistical Manual of Mental Disorders, Fourth Edition, Clinician Version (SCID-CV), administered to subgroups of the total population to assess PPD.				
Freeman et al., 2006 [54]	Omega-3 fatty acid supplement with an EPA:DHA ratio of 1.5:1 in three different dose groups: 0.5 g/day, 1.4 g/day, or 2.8 g/day. Each group received six capsules per day.	21 at the start, 16 remained in the study	Edinburgh Postnatal Depression Scale (EPDS) and Hamilton Rating Scale for Depression (HRSD)		The results showed a significant decrease in depression scores in both groups, with a reduction of 51.5% in the EPDS and 48.8% in the HRSD.	8 weeks	Not mentioned
**Author**	**Findings**		**Authors’ Conclusions**		**Limitations**		
Llorente et al., 2003 [53]	- High ω-3 LC-PUFA intake correlates with reduced depression rates. - Positive relationship between DHA content in plasma and serotonin concentration in the CNS. - DHA capsule group saw reduced PPD symptoms.		- No significant differences in information processing between groups. - No correlations between plasma DHA content and depression scores or Stroop tests.		- Focus on exclusively breastfeeding mothers for 4 months hinders generalization. - Unmeasured factors (e.g., diet) might affect results. - Biological factors, such as genetics, could impact results. - Potential influence from psychological factors like stress or sleep quality.		
Freeman et al., 2006 [54]	- Omega-3 well tolerated with no severe side effects. - EPDS and HRSD scores saw decreases of 51.5% and 48.8%, respectively. - No high-dose advantage over 0.5 g/day.		- No significant pre/post-treatment depression score differences between groups.		- Small sample size and lack of placebo group. - Unequal randomization numbers across groups. - Absence of biochemical measures like pre- and post-treatment essential fatty acid plasma levels.		

**Table 2 healthcare-11-03065-t002:** Differences between prenatal depression and PPD [34,56,57,58,59].

Aspect	Prenatal Depression	PPD
Hormonal Changes	Hyperthyroidism, pituitary adenoma, Cushing’s disease, diabetes mellitus, imbalance in sex steroid hormones, cortisol reactivity, hypothyroxinemia, decreased estradiol levels.	Decreased progesterone levels, decreased estradiol levels, increased testosterone levels, hormonal imbalance, decreased cortisol levels, thyroid dysfunction, sleep disturbances.
Psychological Impact	Has a negative impact on the mother’s mental and physical health, fetal development, and impaired cognitive function.	Impairs mother–infant bonding, deficiencies in the baby’s psychosocial development, negative thoughts about the newborn, affected child’s neurocognitive development, and risk of suicide and infanticide.
Treatment Challenges	Ethical challenges, risk of pharmacological treatment for the fetus, limited studies on the safety of antidepressants during pregnancy.	Dilemmas in treating breastfeeding mothers with medications, concern about the drug passing into breast milk.

**Table 3 healthcare-11-03065-t003:** Characteristics of randomized controlled trials comparing omega-3 treatment with placebo in women with perinatal depression. EPDS: Edinburgh Postnatal Depression Scale. HAM-D: Hamilton Depression Rating Scale. CGI: Clinical Global Impression. HRSD: Hamilton Rating Scale for Depression. MADRS: Montgomery–Åsberg Depression Rating Scale. BDI: Beck Depression Inventory. MINI: Mini International Neuropsychiatric Interview.

Author and Country	Study Title	Design	Type of Participants	Age	Inclusion and Exclusion Criteria
Freeman et al., 2008 [65] (USA)	Omega-3 Fatty Acids and Supportive Psychotherapy for Perinatal Depression: A Randomized Placebo-Controlled Study	Randomized, placebo-controlled	Pregnant women after 12 weeks of gestation	Avg. 30 years	Inclusion: Women 18–45, pregnant or postpartum, MDD diagnosis, EPDS score ≥ 9. Exclusion: Omega-3 intolerance, current medication use, psychosis, bipolar disorder, substance abuse, suicidal ideation.
Rees et al., 2008 [66] (Australia)	Omega-3 fatty acids as a treatment for perinatal depression: randomized double-blind placebo-controlled trial	Randomized controlled trial	Participants were women in the perinatal period	Fish oil: 31.2 years, Placebo: 34.5 years	Inclusion: Pregnant/postpartum women, EPDS ≥ 13, HRSD ≥ 14 or MADRS ≥ 25. Exclusion: Current antidepressants, low depression scores, serious psychiatric/medical history, high omega-3 intake, certain disorders from MINI.
Doornbos et al., 2009 [67] (the Netherlands)	Supplementation of a low dose of DHA or DHA+AA does not prevent peripartum depressive symptoms in a small population-based sample	Randomized controlled trial	Pregnant women in Groningen	Not specified	Inclusion: First or second singleton pregnancies. Exclusion: Vegetarian/vegan diet, diabetes, premature birth.
Mozurkewich et al., 2013 [68] (USA)	The Mothers, Omega-3, and Mental Health Study: a double-blind, randomized controlled trial	Double-blind randomized controlled	Pregnant women in southeastern Michigan	Not specified	Inclusion: Depression history, EPDS 9–19, singleton, 18+ years, 12–20 weeks gestational. Exclusion: Certain disorders, medications, diets, scores from MINI.
Vaz et al., 2017 [18] (Brazil)	Omega-3 supplementation from pregnancy to postpartum to prevent depressive symptoms: a randomized placebo-controlled trial	Randomized, placebo-controlled	Pregnant women in Rio de Janeiro at 20–26 weeks gestation	18 to 40 years	Inclusion: 20–26 weeks gestation, no chronic diseases or psychiatric disorders. Exclusion: Chronic diseases, serious psychiatric disorders, unwillingness for supplements or depression questionnaires.
Farshbaf-Khalili et al., 2017 [69] (Iran)	Fish-Oil Supplementation and Maternal Mental Health: A Triple-Blind, Randomized Controlled Trial	Triple-blind randomized controlled	Pregnant women in Tabriz	Placebo group (26.9 years) and fish oil group (25.9 years)	Inclusion: 18–35 years, 1st–5th pregnancy, EPDS < 20, 16–20 weeks gestational. Exclusion: Certain diseases, disorders, medications, fish or gelatin allergy, high fish intake.
**Author**	**Intervention**		**N**	**Scale Used**	**Duration of Intervention**
Freeman et al., 2008 (USA) [65]	Omega-3 fatty acids + six sessions of supportive psychotherapy		59 pregnant women	EPDS, HAM-D, CGI	Between 12 and 32 weeks of gestation or within six months after childbirth
Rees et al., 2008 (Australia) [66]	Fish oil (2.8 g/day: 1.9 g EPA + 1.1 g DHA)		26 pregnant women	EPDS, HRSD, MADRS	6 weeks
Doornbos et al., 2009 (The Netherlands) [67]	Soybean oil, DHA (220 mg), or DHA+AA (220 mg each) from mid-pregnancy to 3 months post-childbirth		119 pregnant women	EPDS	From week 16 of pregnancy until three months after childbirth
Mozurkewich et al., 2013 (USA) [68]	EPA-rich fish oil, DHA-rich fish oil, or soybean oil placebo		118 pregnant women (divided into 3 groups)	EPDS, BDI, MINI	12–20 weeks of gestation to 6–8 weeks postpartum
Vaz et al., 2017 (Brazil) [18]	Fish oil from week 20 of gestation to 6 weeks post-childbirth		32 women (completed the trial)	EPDS	Week 20 of gestation to 6 weeks post-childbirth
Farshbaf-Khalili et al., 2017 (Iran) [69]	Fish oil capsules (1 g omega-3: 180 mg EPA + 120 mg DHA) or soybean oil placebo		150 pregnant women	EPDS	10 weeks starting at week 16–20 of gestation
**Author**	**Side Effects**		**Findings**		**Limitations**
Freeman et al., 2008 (USA) [65]	Bad breath/unpleasant taste, belching, heartburn/reflux, nausea (22% total; six on omega-3, seven on placebo)		No significant differences between treatment and placebo, but both groups improved.		Small sample size, short treatment duration (8 weeks), potentially low omega-3 dosage.
Rees et al., 2008 (Australia) [66]	Mild reflux, increased stool frequency, nausea		No significant difference, but a trend toward improvement in the fish oil group.		Small sample size, spontaneous remission possibility, no placebo allocation in perinatal period.
Doornbos et al., 2009 (the Netherlands) [67]	Not specified		No prevention of depressive symptoms with DHA or DHA+AA supplementation.		Small sample size due to high dropout, low DHA dosage (220 mg).
Mozurkewich et al., 2013 (USA) [68]	Nausea, belching, and fishy aftertaste		No prevention of depressive symptoms with EPA/DHA. Serum DHA levels predicted BDI scores at 34–36th week.		Lacked statistical power, placebos had fish oil, limited serum DHA measurements, specific population.
Vaz et al., 2017 (Brazil) [23]	Nausea or vomiting, difficulty swallowing capsules		No significant effect, but a trend toward lower postpartum depression in fish oil group.		Placebo effect, small sample size, focus on women with a history of depression.
Farshbaf-Khalili et al., 2017 (Iran) [69]	Nausea, unpleasant taste, vomiting, mild diarrhea, and stomach pain		Reduction in postnatal depression with fish oil. No significant adverse effects on mothers/newborns.		Lack of Beck Depression Scale use, no DHA/EPA umbilical cord level evaluations, potential low omega-3 dosage.

## Data Availability

Not applicable.

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
