# Peer review of "A Critical Look at Omega-3 Supplementation: A Thematic Review"

_healthcare, 2023, doi:10.3390/healthcare11233065_

Round 1

Reviewer 1 Report

Comments and Suggestions for Authors

In the title of the study, the type of review method should be mentioned in detail

In the abstract, details related to the research method and also a summary of the results should be provided

In the research method, information collection tools, how to evaluate studies, and evaluation checklist should be added

The type of review is vague, narrative? Systematic review and...

PRISMA diagram of studies not provided

It is suggested to move tables 1 to 4 after table 7

Results 1 to 4 should be derived from the studies presented in Table 7

Tables 1 to 4 are not related to a review article and mostly represent theoretical foundations

It is suggested that the research hypothesis be carried out according to a strong review method, including a systematic review in detail

Comments on the Quality of English Language

It is suggested that the article be read by a native person in English language

Author Response

Dear reviewer,

We would like to express our sincerest gratitude for taking the time to review and provide valuable feedback on our article titled "Towards a More Effective Treatment for Postpartum Depression: A Critical Look at Omega-3 Supplementation: Review". Your observations and recommendations have been crucial in enhancing the quality and accuracy of the paper.

We are pleased to inform you that we have made the suggested changes and believe that, thanks to your contribution, the article now more accurately and rigorously reflects the research presented.

In the title of the study, the type of review method should be mentioned in detail

Response:

Thank you for pointing out the need to specify the review method in the title. We concur with the importance of this detail. The current work is designed as a thematic review, not a systematic one. The essential difference lies in the methodology of literature collection and analysis. A thematic review focuses on discussing and interpreting various themes or trends based on specific research questions or topics from the selected literature. In contrast, a systematic review adheres to a rigorous methodology with set criteria for article selection and often incorporates a comprehensive meta-analysis. Our decision to opt for a thematic review was to enable a more extensive exploration and amalgamation of varied information sources. Recognizing the importance of clarity in the title, we added Thematic review to the title.

In the abstract, details related to the research method and also a summary of the results should be provided

Response:

It was added to the manuscript.

In the research method, information collection tools, how to evaluate studies, and evaluation checklist should be added

Response:

Our methodology centered around extensive database searches using Boolean algorithms to refine and capture relevant studies. For data storage and selection, we utilized ZOTERO version 6.0.27 (accessed on 20 May 2023) and Rayyan (accessed on 20 May 2023) as our primary tools. I'd like to clarify that our approach was a thematic review, rather than a systematic review. Consequently, we did not apply the quality of evidence or PRISMA guidelines typically associated with systematic reviews. Instead, our thematic review centered on the exploration and interpretation of diverse themes and trends present in the selected literature.

The type of review is vague, narrative? Systematic review and...

Response:

It is a thematic review, we added in the title to clarify.

PRISMA diagram of studies not provided

Response:

Our approach was a thematic review, and as such, we focused on exploring and interpreting various themes or trends within the selected literature, rather than the structured methodology typical of systematic reviews. Given the nature of our review, we did not strictly follow the PRISMA guidelines, which explains the omission of the PRISMA flow diagram. However, we recognize the value of visual representations and will consider incorporating relevant diagrams or charts to enhance clarity in our revised manuscript.

It is suggested to move tables 1 to 4 after table 7

Response:

Tables 1 to 4 were removed, as well as the topics on the definition of depression, to more specifically focus the article on omega-3 as an alternative treatment for postpartum depression

Results 1 to 4 should be derived from the studies presented in Table 7

Response:

Tables 1 to 4 were removed, as well as the topics on the definition of depression, to more specifically focus the article on omega-3 as an alternative treatment for postpartum depression

Tables 1 to 4 are not related to a review article and mostly represent theoretical foundations

Response:

Tables 1 to 4 were removed, as well as the topics on the definition of depression, to more specifically focus the article on omega-3 as an alternative treatment for postpartum depression

It is suggested that the research hypothesis be carried out according to a strong review method, including a systematic review in detail

Response:

We  appreciate this suggestion. However, it's important to clarify that this article is not intended to be a systematic review. The research framework was constructed to move from general theoretical foundations to a more in-depth discussion. While the approach of a systematic review is invaluable, the current structure was chosen to maintain the manuscript's flow and coherence, ensuring a progressive understanding for the readers

Reviewer 2 Report

Comments and Suggestions for Authors

This manuscript describes a narrative review about the efficacy and safety of omega 3 supplementation in postnatal depression. I believe that the overall merit of the manuscript is limited, as its goals seem a little too broad and the order of the information is a little confusing.

Additionally, authors intend to use postpartum depression as the clinical diagnosis for their review. However, I believe that this denomination is too narrow, therefore, the current definition that we use is perinatal depression. Please avoid using interchangeably, the right term is perinatal depression and I it should be used across the whole manuscript.

A flow chart indicating how many records were found when searching the terms and the process of selection should be included, so we can understand hoy many studies are you including.

Please rewrite the description of inclusion and exclusion criteria, as it is now very intricate and hard to understand. Also please note that exclusion criteria are not the opposite of inclusion criteria, e. g. if you already have RCT as an inclusion criterion, please do not mention that non RCT studies will be excluded.

Please also narrow the overall objective of the study. Too much information about postpartum depression was included (pathophysiology and consequences), so narrowing the subject would significantly focus your overall aim.

Authors need to revise scoping review methodology. It seems that this is what is was intended, however, the result was a little messy and unnecessarily wide.

Comments on the Quality of English Language

I believe that the current quality of the manuscript is too low. The current version is poorly organized and the aim is little clear and too broad. I don't believe that a reviewed version of this manuscript could improve it enough to achieve the minimal quality for its publication

Author Response

This manuscript describes a narrative review about the efficacy and safety of omega 3 supplementation in postnatal depression. I believe that the overall merit of the manuscript is limited, as its goals seem a little too broad and the order of the information is a little confusing.

Additionally, authors intend to use postpartum depression as the clinical diagnosis for their review. However, I believe that this denomination is too narrow, therefore, the current definition that we use is perinatal depression. Please avoid using interchangeably, the right term is perinatal depression and I it should be used across the whole manuscript.

Response:

Thank you for your feedback and concerns regarding the terminology and focus of our manuscript. Allow us to clarify and elaborate on why we deem it essential to distinguish between omega 3 supplementation during pregnancy and the postnatal period, as well as the use of the term perinatal.

Physiological Differences Between Pregnancy and the Postnatal Period:

  1. Prenatal Depression: During pregnancy, a woman's body undergoes significant hormonal changes. Hormones like progesterone, estrogen, and cortisol see marked increases. These hormonal fluctuations, impacting both mood and cognitive functions, also influence fatty acid metabolism. For instance, elevated estrogen levels can expedite the conversion of essential fatty acids into EPA and DHA, critical components of omega 3.
  2. Postnatal Depression: Following childbirth, the hormonal landscape shifts dramatically. While levels of certain hormones, such as estrogen and progesterone, drop abruptly, others, like prolactin (crucial for lactation), rise. This postpartum hormonal transition, coupled with physical and psychological factors, can lead to depression. It's pivotal to note that omega 3 supplementation during this period might have a distinct impact compared to its effects during pregnancy due to the differing metabolic and hormonal demands.

Perinatal Terminology: While we understand that perinatal depression is a broad term encompassing both pregnancy and the postnatal period, it's imperative to make clear distinctions. Using this term interchangeably can lead to generalizations that fail to capture the intricacies and specificity of each stage.

Significance of Omega 3 in Neurological Context: The brain, heavily impacted in depressive disorders, has a high concentration of omega 3, notably DHA. During pregnancy, there's an increased demand for DHA to support the fetus's neural development. This demand persists during lactation, as mothers supply DHA to their babies through breast milk. This dynamic could deplete a mother's DHA reserves if a sufficient intake of omega 3 isn't maintained.

A flow chart indicating how many records were found when searching the terms and the process of selection should be included, so we can understand hoy many studies are you including.

Response:

It was done.

Please rewrite the description of inclusion and exclusion criteria, as it is now very intricate and hard to understand. Also please note that exclusion criteria are not the opposite of inclusion criteria, e. g. if you already have RCT as an inclusion criterion, please do not mention that non RCT studies will be excluded.

Response:

It was rewrite.

Please also narrow the overall objective of the study. Too much information about postpartum depression was included (pathophysiology and consequences), so narrowing the subject would significantly focus your overall aim.

Response:

Information was removed to focus the article on what PPD is, its epidemiology, diagnosis, and conventional treatment, and the discussion on the effect of omega 3 supplementation.

Authors need to revise scoping review methodology. It seems that this is what is was intended, however, the result was a little messy and unnecessarily wide.

Response:

We acknowledge the concerns raised about its breadth and organization. As mentioned in a previous comment, we have streamlined the content to specifically focus on the nature of PPD, its epidemiology, diagnosis, conventional treatment, and the implications of omega-3 supplementation. This decision was made to offer a more coherent and targeted narrative.

Reviewer 3 Report

Comments and Suggestions for Authors

The publication submitted for review is interesting, however, it requires improvement. The abstract needs revision. The aim of the study requires refinement. The mechanism of depression should be described in more detail. Additionally, other supplements used in the treatment of depression should be described and compared with omega-3 fatty acids. The authors have presented the results of studies found in the publications, but they have not provided their interpretation. The work is a compilation of data. The conclusion is too brief. Some of the cited publications are unnecessary as they pertain to the same topic of the study and do not bring anything new to the publication. Table 7 requires revision, as it is not very readable. The chapter on diagnosis and conventional treatment needs refinement, as it does not contribute to the sense of the publication, in my opinion. In my view, Chapter 3.9 'Evidence from clinical trials in the treatment of PPD' should be included in the earlier chapter. Chapter 3.7 'Advances in the treatment of PPD' requires revision, as it seems convoluted to me.

Author Response

The publication submitted for review is interesting, however, it requires improvement. The abstract needs revision. The aim of the study requires refinement. The mechanism of depression should be described in more detail. Additionally, other supplements used in the treatment of depression should be described and compared with omega-3 fatty acids. The authors have presented the results of studies found in the publications, but they have not provided their interpretation. The work is a compilation of data. The conclusion is too brief. Some of the cited publications are unnecessary as they pertain to the same topic of the study and do not bring anything new to the publication. Table 7 requires revision, as it is not very readable. The chapter on diagnosis and conventional treatment needs refinement, as it does not contribute to the sense of the publication, in my opinion. In my view, Chapter 3.9 'Evidence from clinical trials in the treatment of PPD' should be included in the earlier chapter. Chapter 3.7 'Advances in the treatment of PPD' requires revision, as it seems convoluted to me.

Response:

Based on the reviewer's comments, We have undertaken the following revisions:

  • The abstract has been revised for clarity and precision.
  • To hone the paper's focus, extraneous content has been eliminated to center primarily on omega-3 supplementation.
  • Detailed interpretations of the compared studies have now been included to offer more than a mere compilation of data.
  • The conclusion section has been expanded and improved to better encapsulate the findings and implications.
  • Table 7 has undergone revisions to enhance its readability and relevance.
  • While the chapter on diagnosis and conventional treatment has been retained, We believe it enriches the narrative by setting a context for the alternative treatment with omega-3. This helps readers understand the broader landscape of treatments before diving into the specifics of omega-3 supplementation.
  • To maintain a focused discussion on omega-3 as a treatment, Chapter 3.7, "Advances in the treatment of PPD," has been removed.

Reviewer 4 Report

Comments and Suggestions for Authors

In the manuscript, Dr. Liscano and Dr. Sanchez-Palacio presented a comprehensive review of postpartum depression (PPD) and various medications to the postpartum depression, especially focus on Omega-3 Supplementation. This review concludes the importance of continuing to explore the relationship between omega-3 supplementation and PPD to enhance the quality of life for pregnant women and their families. In general, the manuscript is well-written in a good organization and in terms of grammar, with a very detailed introduction, sufficient searching and readings of relevant literatures, and very comprehensive presentations of results from different aspects of PPD and treatments. I have some minor concerns are as follows:

1.    For all tables, please include underline to split each row, and shorten texts in each cell. Too many sentences used in tables without splitting them made tables very tedious and are not friendly readable.

2.    For all tables, please add relevant citations to each row or corresponding terms. Without citations, audience are not able to track corresponding articles.

3.    I did not see Section 4 and 5. The authors seem to jump from Section 3 results directly to Section 6 Conclusion. Did the authors omit any sections or it is just a typo of section order?

I really appreciate the way your presented for the detailed Introduction, and the comprehensive review you presented for each aspect of PPD and treatments.

Author Response

Dear reviewer,

We would like to express our sincerest gratitude for taking the time to review and provide valuable feedback on our article titled "Towards a More Effective Treatment for Postpartum Depression: A Critical Look at Omega-3 Supplementation: Review". Your observations and recommendations have been crucial in enhancing the quality and accuracy of the paper.

We are pleased to inform you that we have made the suggested changes and believe that, thanks to your contribution, the article now more accurately and rigorously reflects the research presented.

In the manuscript, Dr. Liscano and Dr. Sanchez-Palacio presented a comprehensive review of postpartum depression (PPD) and various medications to the postpartum depression, especially focus on Omega-3 Supplementation. This review concludes the importance of continuing to explore the relationship between omega-3 supplementation and PPD to enhance the quality of life for pregnant women and their families. In general, the manuscript is well-written in a good organization and in terms of grammar, with a very detailed introduction, sufficient searching and readings of relevant literatures, and very comprehensive presentations of results from different aspects of PPD and treatments. I have some minor concerns are as follows:

  1. For all tables, please include underline to split each row, and shorten texts in each cell. Too many sentences used in tables without splitting them made tables very tedious and are not friendly readable.

Response:

It was corrected.

  1. For all tables, please add relevant citations to each row or corresponding terms. Without citations, audience are not able to track corresponding articles.

Response:

The tables that were omitted were tables 5, 6, and 7, which were renamed respectively as 1, 2, and 3. In particular, references were added to table 2.

  1. I did not see Section 4 and 5. The authors seem to jump from Section 3 results directly to Section 6 Conclusion. Did the authors omit any sections or it is just a typo of section order?

Response:

The sections were corrected from 1 to 4.

I really appreciate the way your presented for the detailed Introduction, and the comprehensive review you presented for each aspect of PPD and treatments.

Reviewer 5 Report

Comments and Suggestions for Authors

The manuscript presents a review of the literature regarding fish oil as a treatment option for postpartum depression.  Implications for the ideas presented in this manuscript have the potential to help mothers, fathers, and babies worldwide. Thank you for bringing focus to this important area of mental health, with a potentially helpful treatment option.

However, the manuscript's current form is a bit too detailed and provides information far broader than what the title suggests.  It is recommended that revisions are made throughout the manuscript to pair down the information and focus specifically on the fish oil treatment option.  Background information regarding depression and the various forms should be limited.  Review of the literature regarding depression should be summarized more succinctly.  

Below are a few specific recommendations for revision:

Introduction: There are numerous claims stated without proper citation.

Materials and Methods:  It is not necessary to include the search terms.

Results: Tables 1 & 2 should be reformatted so they are more succinct and easier to read.  Table 5 should either be a) revised toward more succinct and clear comparisons or b) omitted and replaced by a concise descriptive paragraph.  Table 7 is too long and detailed; should be revised to be more succinct.

Author Response

Dear reviewer,

We would like to express our sincerest gratitude for taking the time to review and provide valuable feedback on our article titled "Towards a More Effective Treatment for Postpartum Depression: A Critical Look at Omega-3 Supplementation: Review". Your observations and recommendations have been crucial in enhancing the quality and accuracy of the paper.

We are pleased to inform you that we have made the suggested changes and believe that, thanks to your contribution, the article now more accurately and rigorously reflects the research presented.

The manuscript presents a review of the literature regarding fish oil as a treatment option for postpartum depression.  Implications for the ideas presented in this manuscript have the potential to help mothers, fathers, and babies worldwide. Thank you for bringing focus to this important area of mental health, with a potentially helpful treatment option.

However, the manuscript's current form is a bit too detailed and provides information far broader than what the title suggests.  It is recommended that revisions are made throughout the manuscript to pair down the information and focus specifically on the fish oil treatment option.  Background information regarding depression and the various forms should be limited.  Review of the literature regarding depression should be summarized more succinctly.  

Below are a few specific recommendations for revision:

Introduction: There are numerous claims stated without proper citation.

Response:

It was corrected.

Materials and Methods:  It is not necessary to include the search terms.

Response:

Given that we conducted a thematic (not systematic) review rather than a narrative one, we employed a methodology to identify articles on postpartum depression using specific inclusion and exclusion criteria

Results: Tables 1 & 2 should be reformatted so they are more succinct and easier to read.  Table 5 should either be a) revised toward more succinct and clear comparisons or b) omitted and replaced by a concise descriptive paragraph.  Table 7 is too long and detailed; should be revised to be more succinct.

Response:

It was corrected.

Round 2

Reviewer 1 Report

Comments and Suggestions for Authors

Respected Authors

Thank you for your responses. You have answered almost all the comments, but still, a few issues are not clear, which are stated below. 

- Title, ":" has been repeated twice in your title. This method of writing is not scientific and according to the guidelines of the journal. Please remove one of them and refine your title. As your results are not similar enough, then we can't say that this supplement is effective. Then, my suggested title is as follows: "A Critical Look at Omega-3 Supplementation for Treatment of Postpartum Depression: A Thematic Review"

- Abstract, the way you reported the abstract is not scientific. Please remove all study authors from this section. 

- Please remove the search strategy from the main text and insert it as a supplementary file.

- Please remove bullet points from section 2.2.3. Please rewrite it as a paragraph. 

- In revision, you stated that this is a thematic review but there is still not a synthesis section in your methods. Please add the way you synthesize or categorize your results. 

- Please remove references from the table caption. 

Author Response

Dear Reviewer,

We sincerely appreciate your valuable comments and suggestions for improving our article. Your observations are highly valuable to us and enable us to enhance our work.

Reviewer: Thank you for your responses. You have answered almost all the comments, but still, a few issues are not clear, which are stated below. 

- Title, ":" has been repeated twice in your title. This method of writing is not scientific and according to the guidelines of the journal. Please remove one of them and refine your title. As your results are not similar enough, then we can't say that this supplement is effective. Then, my suggested title is as follows: "A Critical Look at Omega-3 Supplementation for Treatment of Postpartum Depression: A Thematic Review"

Response:

It was modified.

- Abstract, the way you reported the abstract is not scientific. Please remove all study authors from this section. 

Response:

It was modified.

- Please remove the search strategy from the main text and insert it as a supplementary file.

Response:

It was modified.

- Please remove bullet points from section 2.2.3. Please rewrite it as a paragraph. 

Response:

It was modified.

- In revision, you stated that this is a thematic review but there is still not a synthesis section in your methods. Please add the way you synthesize or categorize your results. 

Response:

We appreciate your comments and suggestions to improve our article. However, we would like to clarify that our methodology is based on a thematic review rather than a thematic synthesis. We understand the fundamental difference between these two approaches and believe that our methodology is specifically designed to address the objectives and scope of our review.

Regarding the request for detailed information about the synthesis process, we believe this may be unnecessary in the context of a thematic review. Our primary intention is to provide a comprehensive and understandable review of the existing literature on the subject in question, focusing on the identification and analysis of key themes and trends rather than conducting a formal synthesis of qualitative data.

- Please remove references from the table caption. 

Response: We appreciate the comment; however, the inclusion of the caption had been previously suggested by another reviewer

Reviewer 3 Report

Comments and Suggestions for Authors

 Accept in present form

Author Response

Dear Reviewer,

We sincerely appreciate your valuable comments and suggestions for improving our article. Your observations are highly valuable to us and enable us to enhance our work.